# Retinal Microvascular Changes in Familial Hypercholesterolemia: Analysis with Swept-Source Optical Coherence Tomography Angiography

**DOI:** 10.3390/jpm12060871

**Published:** 2022-05-26

**Authors:** Pétra Eid, Louis Arnould, Pierre-Henry Gabrielle, Ludwig S. Aho, Michel Farnier, Catherine Creuzot-Garcher, Yves Cottin

**Affiliations:** 1Ophthalmology Department, University Hospital, 21000 Dijon, France; petra.eid@chu-dijon.fr (P.E.); louis.arnould@chu-dijon.fr (L.A.); phgabrielle@gmail.com (P.-H.G.); 2INSERM, CIC1432, Clinical Epidemiology Unit, Dijon University Hospital, 21000 Dijon, France; 3Centre des Sciences du Gout et de l’Alimentation, AgroSup Dijon, CNRS, INRAE, University of Burgundy Franche-Comté, 21000 Dijon, France; 4Epidemiology Department, University Hospital, 21000 Dijon, France; ludwig.aho@chu-dijon.fr; 5Lipid Clinic, Point Medical and Department of Cardiology, University Hospital, 21000 Dijon, France; farnier.michel@orange.fr; 6Cardiology Department, University Hospital, 21000 Dijon, France; yves.cottin@chu-dijon.fr; 7PEC 2, University Bourgogne Franche-Comte, 21000 Dijon, France

**Keywords:** swept-source optical coherence tomography angiography, familial hypercholesterolemia, Coronary Artery Calcium score, foveal avascular zone, retinal vascular densities

## Abstract

Familial hypercholesterolemia (FH) is a common but underdiagnosed genetic disorder affecting cholesterol metabolism, leading to atherosclerotic disease. The relationship between retinal microvascular changes and the presence of atheroma in patients with FH (FH group), and in comparison to volunteers without FH (CT group), needs further investigation. This cross-sectional study was conducted in a university hospital between October 1, 2020 and May 31, 2021. Cardiovascular data, including the Coronary Artery Calcium (CAC) score, were recorded for FH patients. Macula angiograms were acquired using swept-source optical coherence tomography angiography (SS OCT-A) to analyze both the superficial capillary plexus (SCP) and deep capillary plexus (DCP). A total of 162 eyes of 83 patients were enrolled in the FH group and 121 eyes of 78 volunteers in the CT group. A statistically significant association was found between the CAC score and both vessel density (β = −0.002 [95% CI, −0.004; −0.0005], *p* = 0.010) and vessel length (β = −0.00005 [95% CI, −0.00008; −0.00001], *p* = 0.010) in the DCP. The FH group had a significantly lower foveal avascular zone circularity index than the CT group in multivariate analysis (0.67 ± 0.16 in the FH group vs. 0.72 ± 0.10 in the CT group, β = 0.04 [95% CI, 0.002; 0.07], *p* = 0.037). Retinal microvascularization is altered in FH and retinal vascular densities are modified according to the CAC score.

## 1. Introduction

Familial hypercholesterolemia (FH) is a common genetic disorder affecting 1 in 500 births in France [1,2]. Due to a mutation in genes involved in the metabolism of low-density lipoprotein cholesterol (LDL-c), FH is characterized by a significant elevation of LDL-c from birth that persists throughout life [3]. In the most frequent form, characterized by an autosomal dominant heterozygous transmission, cardiovascular risk is increased 13-fold compared to the general population [4]. While early identification and adequate treatment can improve the prognosis of this congenital disease, the diagnosis of FH is often made late after a major cardiovascular event occurs [5]. In most cases, the diagnosis of FH is based on clinical presentation. The Dutch Lipid Clinic Network Score is a validated set of criteria based on familial history of premature cardiovascular disease, patient history of premature cardiovascular disease, physical signs, untreated LDL-c levels, and the genotype. According to this score, the diagnosis of FH ranges from possible to definite [3,6]. FH is marked by high phenotype variability and heterogeneity in the atherosclerotic course for the same LDL-c levels. Common coronary heart disease risk equations (the Framingham Risk Score) failed to determine precisely the individual cardiovascular prognosis in this patient population [7]. The Coronary Artery Calcium (CAC) score, which emerged over the past few decades, provides a subclinical measure of coronary atherosclerosis using computed tomography [8]. The CAC score offers a more personalized risk assessment in cases of FH and helps to guide the treatment decisions [9,10].

High LDL-c levels in FH are associated with some ocular manifestations, such as corneal arcus, which is a diagnosis criterion for FH in the Dutch Lipid Clinic Network Score when it appears prior to 45 years of age [11]. However, no characteristic manifestations of hypercholesterolemia have been described at the retinal level. Swept-source optical coherence tomography angiography (SS OCT-A) is a fast and noninvasive retinal imaging technology that enables three-dimensional visualization of the retinal microvascularization and foveal avascular zone (FAZ). The machine’s software provides a quantitative analysis of the superficial and deep retinal capillary plexuses (SCP and DCP, respectively) [12]. Used in daily clinical practice for the diagnosis and follow-up of retinal vascular diseases, this examination also facilitates the investigation of OCTA-based microvascular metrics [13]. These quantitative parameters describe early retinal microvascular changes in systemic pathologies. Numerous previous studies found associations between abnormal OCTA-based retinal metrics and diabetes, neurodegenerative disease, and cardiovascular events [14,15,16,17].

We hypothesized that SS OCT-A quantitative parameters could help to elucidate cardiovascular risk stratification in FH patients. Moreover, SS OCT-A could represent an additional tool with which to refine risk estimates when using the CAC score for these patients. 

This study aimed to (a) analyze the association between retinal vascular density and the presence of atherosclerosis assessed by the CAC score and (b) compare SS OCT-A quantitative parameters between patients with FH and volunteers without a history of FH.

## 2. Materials and Methods

### 2.1. Participants

This cross-sectional observational study was conducted in the Ophthalmology and Cardiology departments of Dijon University Hospital. We included consecutive patients diagnosed with FH who attended a follow-up cardiology visit between 1 October 2020 and 31 May 2021 (FH group). The number of patients included in this pilot study was based on the capacity of recruitment in the Cardiology department relying on the rhythm of FH patients’ follow-up. The diagnosis of FH was based on the Dutch Lipid Clinic Network criteria. Demographic data, medical history, and cardiovascular data, including the CAC score, were collected from cardiology medical records. Volunteers with no history of FH were recruited in the Ophthalmology department during the same period to constitute the control group (CT group) in order to achieve a 1:1 ratio. Volunteers were patients consulting the Ophthalmology department for various purposes, excluding patients with retinal diseases. 

The exclusion criteria were eyes with a history of retinal disease, optic neuropathy, or a poor-quality image in SS OCT-A (signal strength of ≤7/10 or presence of artifacts due to eye movement or media opacities). The study adhered to the tenets of the Declaration of Helsinki and followed the STROBE statements [18]; approval from the local ethics committee was obtained (no. 2017-A02724-49). Written informed consent was obtained for each patient after explaining the nature and possible consequences of the study.

### 2.2. Imaging

The study was conducted with 6 × 6 mm^2^ macular angiograms acquired with an SS OCT-A (PLEX Elite 9000 Swept-Source, Carl Zeiss Meditec, Inc., Dublin, Ireland) device, and only images with a signal strength of >7/10 and without significant artifacts were retained. Quantitative SS OCT-A data were acquired using the Macular Density v0.7.3 algorithm available online through the ARI Network (Carl Zeiss Meditec, Inc., Dublin, Ireland) (Figure 1). This includes a projection-artifact-removing algorithm that allows for a more refined analysis of the DCP. In addition to SS OCT-A, we collected information on objective refraction, intraocular pressure (Nidek Tonoref II autorefractokeratometer, Nidek Ltd., Gamagori, Japan), axial length (IOL Master 500, Carl Zeiss Meditec Inc., Dublin, USA), slit-lamp examination results, and ultra-widefield fundus imaging findings (Clarus 500, Carl Zeiss Meditec Inc., Dublin, Ireland and/or Optos California, Optos Inc., Malborough, UK). Slit-lamp examination allowed us to analyze the anterior segment including the lens status, and fundus photography was checked to assess the absence of retinal disease and optic neuropathy. 

The study of quantitative retinal microvascularization was based on the analysis of vessel density (given in mm^−1^) and vessel length (ratio, unitless) in both the SCP and DCP. The area size, perimeter, and circularity index of a specific retinal zone devoid of vessels in the SCP, the foveal avascular zone (FAZ size given in mm^2^, FAZ length in mm, and FAZ circularity index unitless) were also studied. All quantitative SS OCT-A data were obtained using the Macular Density v0.7.3 algorithm available online through the ARI Network (Carl Zeiss Meditec, Inc., Dublin, Ireland).

### 2.3. Coronary Artery Calcium Score

The CAC score is a highly specific instrument used to stratify cardiovascular risk [19]. Determination of the CAC score is based on analysis of axial CT scans of the heart. Briefly, detecting a lesion over 130 HU with an area greater than 1 mm^2^ is considered a calcified area in a given coronary artery. The measured density is stratified and multiplied by the area of the coronary calcification. The calcium score of each coronary artery is then summed to obtain the total CAC score [8].

### 2.4. Statistical Analysis 

Statistical analysis was performed with STATA software (StataCorp, version 15.1, LLC, TX, USA). Continuous variables are expressed as medians (interquartile range). Categorical variables are expressed as numbers (percentage). The chi-square test or Fisher’s exact test was used to compare categorical variables, and the Mann–Whitney test was used to compare quantitative data. A linear mixed model was used to consider the non-independence between both eyes for one patient [20]. A multivariate model adjusted for age, gender, hypertension, and smoking status was used to analyze retinal microvascularization according to the CAC score in FH patients and to compare the FH and CT groups. Diabetes was not included in our multivariate model because of the small number of patients with diabetes in both groups. Analyses were performed with a robust variance estimator. Linearity was assessed using a fractional polynomial. A two-tailed *p* value of < 0.05 was considered statistically significant.

## 3. Results

The study comprised a total of 283 eyes of 161 participants, including 162 eyes of 83 patients affected with FH and 121 eyes of 78 volunteers without a known personal or familial history of FH and not taking cholesterol-lowering medication.

### 3.1. Characteristics of the Study Population

Eighty-nine FH patients were screened for this study; two patients refused to participate and four were excluded due to retinal disease or impossibility to achieve high-quality SS OCT-A acquisitions (lack of fixation, advanced cataract).

As presented in Table 1, the demographic characteristics of patients from the FH and CT groups did not show any statistically significant difference. All patients from the FH group had a diagnosis ranging from probable to definite according to the Dutch Lipid Clinic Network Score (11 patients had a probable diagnosis with a score between 6 and 8 points, and 72 patients had a definite diagnosis with a score over 8 points, mainly due to an identified genetic mutation). Most FH patients had a heterozygous mutation in the LDL-receptor gene (80.6%), the most common cause of FH. Systemic hypertension was more prevalent in the FH group than in the CT group (36.1% in the FH group versus 14.5% in CT group, *p* = 0.002), whereas the rate of diabetes was similar in both groups. FH patients had a higher rate of exposure to smoking than the CT patients did (43.4% of current or former smokers in the FH group vs. 18.7% in the CT group, *p* < 0.001). FH patients presented various carotid artery conditions. Among the 83 patients from FH group, data concerning carotid artery disease were available for 74 patients (10.8% missing data). Fifty-eight patients (69.9%) were free of significant carotid artery stenosis and 16 patients (19.3%) presented with a significant carotid stenosis or a history of carotid artery stenting or endarterectomy. More patients in the control group than in the FH group had cataract surgery (2.5% pseudophakia in the FH group vs. 11.6% in the CT group, *p* = 0.002). 

### 3.2. Retinal Microvascularization and CAC Score in the FH Group 

Table 2 shows the association between SS OCT-A parameters and CAC score. The CAC score was available for 55 patients (109 eyes). In a multivariate model adjusted for age, gender, hypertension, and former or current smoking status, a statistically significant association was found between CAC score and vascular densities in the DCP (β = −0.002 [95% CI, −0.004; −0.0005], *P* = 0.010 for vessel density in the DCP and β = −0.00005 [95% CI, −0.00008; −0.00001], *p* = 0.010 for vessel length in the DCP). No statistically significant association was found for vascular densities in the SCP, nor for FAZ parameters. Age appeared to be a confounding factor for vessel length in the SCP (β = 0.0005 [95% CI, 0.001; 0.00005], *p* = 0.030) and for vessel density in the DCP (β = 0.06 [95% CI, 0.11; 0.008], *p* = 0.025). 

### 3.3. Retinal Microvascularization in the FH and CT Groups

Table 3 shows the comparison between retinal vascular parameters in the FH and CT groups. After adjusting for age, gender, hypertension, and smoking status, there was no statistically significant difference between the FH and CT groups for vessel density and vessel length in both the SCP and DCP. The FH group was associated with a statistically significantly lower FAZ circularity index than the CT group in the multivariate analysis (0.67 ± 0.16 in the FH group vs. 0.72 ± 0.10 in the CT group, β = 0.04 [95% CI, 0.002; 0.07], *p* = 0.037). There was no significant difference between the two groups for FAZ size or FAZ length. Age was the only confounding factor between the FH and CT groups associated with a significant modification of FAZ length (β = 0.02 [95% CI, 0.002; 0.030], *p* = 0.030), FAZ size (β = −0.002 [95% CI, 0.0004; 0.005], *p* = 0.025), and FAZ circularity (β = −0.002 [95% CI, −0.003; −0.0008], *p* <0.0001). For vessel length in the SCP and vessel density in the DCP, both age (β = −0.0004 [95% CI, −0.0008; −0.00008], *p* = 0.014 and β = −0.06 [95% CI, −0.09; −0.03], *p* <0.0001, respectively) and hypertension (β = −0.02 [95% CI, −0.03; −0.001], *p* = 0.034 and β = −1.50 [95% CI, −2.85; −0.15], *p* = 0.030, respectively) were confounding factors.

## 4. Discussion

To our knowledge, this is the first study to describe the relationship between changes in retinal microcirculation and systemic vascular parameters in FH. We highlighted the potential role of quantitative SS OCT-A as a biomarker of atherosclerosis evaluated with the CAC score in FH patients. 

In FH, chronic exposure to high levels of LDL-c leads to the early development of atherosclerosis, which represents a turning point in developing cardiovascular complications [6]. Due to vascular dysfunction, arteriolar remodeling and capillary rarefaction are early and well-established phenomena observed in hypercholesterolemia that may play a role in various types of organ dysfunction, such as kidney and heart failure [21]. The eye is a unique organ that offers direct in vivo visual access to the microvasculature. Many studies have focused on retinal microvascular modifications by exploring retinal endothelial dysfunction in hypercholesterolemia with experimental devices [22,23]. As shown in our study, this microvascular rarefaction can be precisely quantified in the retinal capillary plexuses using SS OCT-A imaging. At the same time, SS OCT-A is a noninvasive, non-irradiant, low-cost, easily accessible, and daily-use technology for the in vivo assessment of retinal microvascularization. Furthermore, SS OCT-A provides accurate, objective, reproducible, and automated analysis of retinal microvascularization, offering immediate results. We could envision in the future the use of SS OCT-A during cardiological consultations as a supplementary tool when monitoring atherosclerosis in FH patients. In practical terms, a lowering in vascular densities could indicate an atherosclerosis burden and lead to further cardiological investigation and/or adjustment of LDL-c-lowering medication. 

The commonly used risk assessment tools and risk calculators have low predictability and underestimate cardiovascular risk in FH. The CAC score represents a more recent relevant tool in coronary heart disease, atherosclerotic cardiovascular disease, and mortality risk prediction in FH. However, it requires cardiac computed tomography, an expensive, not readily available, and irradiating imaging examination [24,25]. As a window on the systemic vascular state, retinal vascular damage and cardiovascular status have a well-known and extensively studied link [26,27]. SS OCT-A represents a supplementary examination that can offer a more personalized evaluation of systemic vascular status, especially if SS OCT-A biomarkers precede systemic complications, as observed in diabetic retinopathy [28,29]. In further longitudinal studies, it would be interesting to evaluate SS OCT-A as a predictive tool of systemic complications in FH. Moreover, management of FH patients is currently based on the control of blood LDL-c levels. However, LDL-c measurement in a blood sample at a given time does not reflect previous months of cholesterol balance or imbalance. Nonetheless, it is the chronic exposure to high LDL-c levels that leads to cardiovascular disease. A longitudinal study could provide further information about changes in retinal microvascularization over time and the potential memory of retinal vascular densities as a reflection of previous months’ LDL-c levels. 

This study found a lower FAZ circularity index in the FH group than in the CT group in multivariate analysis adjusting for general cardiovascular risk factors. Thus far, this is one of the first studies to describe an association between modifications in the FAZ architecture and systemic vascular changes. Since this mechanism is poorly documented, our hypothesis relies on the phenomenon of microvascular focal ischemia and perifoveal microvascular irregular remodeling. However, the comparison of our results with other studies is limited. Indeed, there are numerous OCT-A devices with different segmentations of the retinal and vascular layers. There are also many different algorithms used to obtain quantitative data on retinal microvascularization. In a multivariate model adjusted for age, gender, hypertension, and former or current smoking status, a statistically significant association was found between CAC score and vascular densities and vessel length in the DCP. In this study, the association between SS OCT-A parameters and CAC score was only found in the DCP. A similar modification of retinal microvascularization, more pronounced in the DCP, was described in cases of hypertension and erectile dysfunction [30,31]. In acute coronary syndrome and in acute renal injury after acute coronary syndrome, changes in retinal microvascularization were found only in the SCP, but no conclusions regarding the DCP were available since the DCP was not analyzed [14,32]. The DCP has a lower vascular density and vessel length than the SCP, making it more sensitive to systemic modifications [33]. As found in this study, age is a common confounding factor described in most studies focusing on OCT-A and systemic disease. However, numerous other systemic factors influence retinal microvascularization, possibly in a different way between the SCP and DCP [34,35,36]. Accurate studies of the association between systemic parameters and retinal microvascularization require that the most detailed information on cardiovascular risk factors, general medical history, and treatments is provided. 

Recently, a large study highlighted the value of retinal photograph-based deep learning as an alternative measure of CAC score and for the prediction of cardiovascular events [37,38]. However, to our knowledge, there is no study to date analyzing OCT-A-derived retinal microvascular measures and the CAC score. This preliminary approach opens the way to assessing the value of OCT-A-based artificial intelligence analysis for CAC score evaluation. The quantitative analysis of retinal vascular density could implement automatic algorithms in order to refine cardiovascular risk stratification [37,38,39]. A quantitative analysis of retinal microvascularization using SS OCT-A imaging could offer an interesting new perspective on the link between the retina and atherosclerosis evaluation in FH.

The main limitation of this study is its cross-sectional design, which prevents us from drawing definitive conclusions with a longitudinal follow-up of retinal vascular densities and cardiovascular status. Further longitudinal evaluations are warranted to confirm our findings. Another point to note is the relatively small number of patients in the FH group, considering the frequency of this disease. This point is directly related to the misdiagnosis of FH in the general population [4]. In addition to this, exclusion criteria were based only on the absence of ophthalmological disease, while other general affectations such as neurological disorders could also impact retinal vascular densities [40]. Concerning the volunteers constituting the CT group, they were recruited on the basis of a self-reported absence of personal or familial FH history and not being on cholesterol-lowering medications. Since FH is a largely underdiagnosed disease, the absence of an unknown form of FH in the CT group cannot be established with certainty. Therefore, screening for lipid anomalies in the CT group should be considered in further studies. Furthermore, almost one third of the patients in the FH group did not undergo CAC score assessment. This represents a potential bias since the CAC score is preferentially evaluated to detect subclinical atherosclerosis in primary prevention [10]. A systematic evaluation of the CAC score in all FH patients in a future study would make it possible to evaluate a potential differential relationship between the CAC score and retinal microvascularization in OCT-A depending on the presence or absence of systemic complications. Moreover, this study’s single-center design, which was conducted in a tertiary referral center, may have led to the selection of FH patients with more severe or complex presentations of FH. Finally, there was a statistically significant difference in ocular characteristics between the two groups, with more phakic patients found in the FH group. Since lens status may be associated with lens opacity, this could have an impact on OCT-A analysis by altering the retinal microvascular quantification. Indeed, numerous artifacts can affect the quantitative and qualitative interpretation of OCT-A [41]. However, the careful selection of OCT-A images with a signal strength of > 7/10 and no significant artifacts permitted us to obtain similar image quality between the two groups. This attenuated the consequence of lens status differences between the groups. This point highlights the importance of strict quality control when it comes to microvascularization analysis with OCT-A.

## 5. Conclusions

Retinal vascular density assessed with SS OCT-A, as a window to systemic vascular modifications, could represent an innovative approach for detecting early microvascular modifications and investigating atherosclerosis. The relationship between OCT-A and CAC score indirectly reflects the cardiovascular risk profile in patients with FH, but further studies are necessary to evaluate SS OCT-A as a cardiovascular risk profile biomarker in FH. 

## Figures and Tables

**Figure 1 jpm-12-00871-f001:**
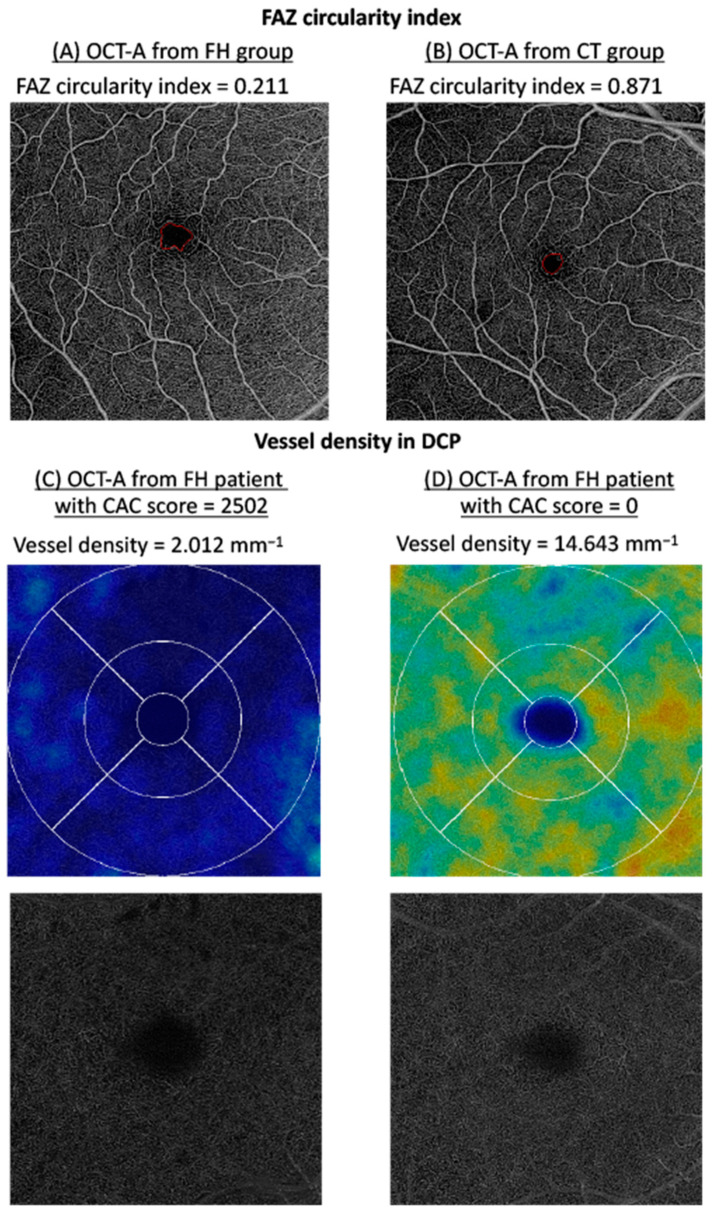
Examples of OCT-A images. Illustration of FAZ circularity index in FH group (**A**) and CT group (**B**) and vessel density in the DCP in FH patients with a high CAC score (**C**) and a low CAC score (**D**). FAZ: foveal avascular zone. OCT-A: optical coherence tomography angiography. FH group: familial hypercholesterolemia group. CT group: control group. CAC score: Coronary Artery Calcium score. mm: millimeter. mm^−1^: per millimeter.

**Table 1 jpm-12-00871-t001:** Comparison of patient characteristics between familial hypercholesterolemia and control group.

	FH Group	CT Group	*p*
Patients (eyes)	83 (162)	78 (121)	
Age (years)	56.0 (41.0; 69.0)	49.1 (30.0; 66.0)	0.342
Gender (male)	44 (53.0)	31 (39.7)	0.092
Cardiovascular history			
Hypertension	30 (36.1)	11 (14.5)	**0.002**
Diabetes	4 (4.8)	3 (4.2)	0.860
Smoking status (current of former smoker)	36 (43.4)	14 (18.7)	**<0.001**
Ocular characteristics			
Intraocular pressure (mmHg)	15.0 (13.0; 18.0)	15.0 (13.0; 18.0)	0.524
Axial length (mm)	23.5 (22.9; 24.0)	23.4 (23.0; 24.3)	0.659
Lens status (phakic)	158 (97.5)	107 (88.4)	**0.002**

FH group: familial hypercholesterolemia group. CT group: control group. mmHg: millimeter of mercury. mm: millimeter. Results are presented as n (%) and median (interquartile range). Chi-square test was used to compare categorical variables and Mann–Whitney test was used to compare quantitative data. Significant *p* values are highlighted in bold.

**Table 2 jpm-12-00871-t002:** Association between retinal vascular parameters and Coronary Artery Calcium score in familial hypercholesterolemia group eyes.

	Multivariate Analysis (*n* = 109)	
	β [95% CI]	*p*
Superficial capillary plexus		
Vessel density (mm^−1^)	−0.0009 [−0.002; 0.0003]	0.162
Vessel length	−0.00002 [−0.0003; 0.0000008]	0.067 *
Deep capillary plexus		
Vessel density (mm^−1^)	−0.002 [−0.004; −0.0005]	**0.010** *
Vessel length	−0.00005 [−0.00008; −0.00001]	**0.010**
FAZ size (mm^2^)	0.0001 [−0.00002; 0.0003]	0.101
FAZ length (mm)	−0.0003 [−0.0001; 0.001]	0.117
FAZ circularity index	−0.00002 [−0.0008; −0.00005]	0.573

FH group: familial hypercholesterolemia group. β: regression coefficient. CI: confidence interval. FAZ: foveal avascular zone. mm: millimeter. mm^−1^: per millimeter. mm^2^: square millimeter. Multivariate mixed model regression adjusted for age, gender, hypertension, and current or former smoking status was used to compare quantitative data. * Age is a confounding factor. Significant *p* values are highlighted in bold.

**Table 3 jpm-12-00871-t003:** Comparison of retinal vascular parameters between familial hypercholesterolemia and control group eyes.

			Multivariate Analysis	
	FH Group (*n* = 162)	CT Group (*n* = 121)	β [95% CI]	*p*
Superficial capillary plexus				
Vessel density (mm^−1^)	18.73 ± 2.41	19.34 ± 1.51	0.38 [−0.22; 0.07]	0.211
Vessel length	0.41 ± 0.05	0.42 ± 0.03	0.007 [−0.006; 0.02]	0.297 *^†^
Deep capillary plexus				
Vessel density (mm^−1^)	11.26 ± 4.69	12.14 ± 4.04	0.46 [−0.66; 1.58]	0.419 *^†^
Vessel length	0.26 ± 0.47	0.24 ± 0.08	−0.003 [−0.09; 0.09]	0.944
FAZ size (mm^2^)	0.32 ± 0.47	0.27 ± 0.16	−0.02 [−1.11; 0.08]	0.701 *
FAZ length (mm)	2.45 ± 2.63	2.13 ± 0.77	−0.13 [0.65; 0.39]	0.627 *
FAZ circularity index	0.67 ± 0.16	0.72 ± 0.10	0.04 [0.002; 0.07]	**0.037 ***

FH group: familial hypercholesterolemia group. CT group: control group. β: regression coefficient. CI: confidence interval. FAZ: foveal avascular zone. mm: millimeter. mm^−1^: per millimeter. mm^2^: square millimeter. Descriptive results for FH and CT groups are presented as mean ± standard deviation. Multivariate mixed model regression adjusted for age, gender, hypertension, and current or former smoking status was used to compare quantitative data. * Age is a confounding factor. ^†^ Hypertension is a confounding factor. Significant *p* value is highlighted in bold.

## Data Availability

Presented data are available on request from the corresponding author.

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
