# Peer review of "Retinal Microvascular Changes in Familial Hypercholesterolemia: Analysis with Swept-Source Optical Coherence Tomography Angiography"

_jpm, 2022, doi:10.3390/jpm12060871_

Round 1

Reviewer 1 Report

This study investigated the retinal microvascular changes in patients with familial hypercholesterolemia (FH). A total of 162 eyes from 83 patients and 121 eyes from 78 volunteers were included in the study. cardiovascular data and macular optical coherence tomography angiography (OCTA) data were collected. The authors found that the FH group had a significantly lower foveal avascular zone circularity index. The authors conclude that retinal microvascularization is altered in FH patients.

Do you have any data on the condition of the carotid artery in these patients? Any carotid plaques or stenosis?

Author Response

Manuscript ID jpm-1694763

“Retinal microvascular changes in familial hypercholesterolemia: 
analysis with swept-source optical coherence tomography angiography”

Thank you for giving us the opportunity to revise this submission following peer review. And many thanks for your thorough review, your comments are highly appreciated and very helpful for us. After consultation with the other authors, we have responded to the reviewer’s constructive suggestions by revising the manuscript.

Please find below our response to the Reviewer’s suggestions.

Reviewer 1: This study investigated the retinal microvascular changes in patients with familial hypercholesterolemia (FH). A total of 162 eyes from 83 patients and 121 eyes from 78 volunteers were included in the study. cardiovascular data and macular optical coherence tomography angiography (OCTA) data were collected. The authors found that the FH group had a significantly lower foveal avascular zone circularity index. The authors conclude that retinal microvascularization is altered in FH patients.

Do you have any data on the condition of the carotid artery in these patients? Any carotid plaques or stenosis?

We agree with this concern raised by the Reviewer. Indeed, we did not presented data on the condition of the carotid artery in FH patients but we do have collected this information. In our FH population, 58 patients (69.9%) were free of significant carotid artery stenosis, 16 patients (19.3%) had carotid artery plaque with significant stenosis or a history of carotid artery stenting or endarterectomy. Significant carotid artery stenosis corresponded to a symptomatic carotid artery disease or a narrowing of the arterial lumen of more than 60%. This data was missing for 9 patients (10.8%).

We added the following statement in the revised version of our manuscript (line 167-172)

“FH patients presented various carotid artery conditions. Among the 83 patients from FH group, data concerning carotid artery disease was available for 74 patients (10.8% missing data). Fifty-eight patients (69.9%) were free of significant carotid artery stenosis and 16 patients (19.3%) presented with a significant carotid stenosis or a history of carotid artery stenting or endarterectomy.”

Reviewer 2 Report

The authors presented an interesting study investigating the relationship between familial hypercholesterolemia and retinal microvascular changes assessed by optical coherence tomography angiography. The topic of the manuscript is original in content, and the findings are worth reporting, but the authors should improve the manuscript for English language and address the following comments before publication can be considered.

Abstract

  • Lines 19-21: please reformulate the sentence not as a question but in a more scientific way, such as “It is not known if an association exists between retinal microvascular changes and the presence of atheroma in patients with FH” or “the relationship between FH and retinal microvascular changes needs further investigation” or a similar statement to introduce the rationale fro the study, but not in a form of a question

Introduction

  • The manuscript should be revised and improved for English language: for example at line 14 “In the most frequent form, autosomal dominant heterozygous transmission, cardiovascular risk is increased 13-fold compared to the general population [4].” It should be replaced with “In the most frequent form characterized by/with an autosomal dominant heterozygous transmission, cardiovascular risk is increased 13-fold compared to the general population [4].”
  • Before the paragraph related to OCTA, the authors should add some insight regarding FH and the eye. Has FH any known ocular clinical manifestations?

Methods

  • Line 97-98: “on objective refraction, intraocular pressure, axial length, slit-lamp examination results”: the authors should provide additional information about the instruments used to measure “objective refraction, intraocular pressure, axial length” and indicate the details of what they mean by “slit-lamp examination results”
  • Statistics: The author should provide a statistical power estimation for their study or at least some justification of the study n and add it to the “statistical analysis section

Results

  • Line 142-143: “As presented in Table 1, the demographic characteristics of patients from the FH and CT groups were similar: please replace the term “similar” with a more objective and scientific term, or reformulate the sentence.
  • When commenting about the differences between the two groups in terms of demographic characteristics, the authors should add in the description the indication of the statistical significance (p-value) of the differences, and not only the percentages.

Discussion

  • All the limitations of the study should be presented in a single paragraph, and not in multiple ones at the end of the discussion
  • The authors should provide more insight about the potential practical applicability in the clinical practice of their findings

Author Response

Manuscript ID jpm-1694763

“Retinal microvascular changes in familial hypercholesterolemia: 
analysis with swept-source optical coherence tomography angiography”

Thank you for giving us the opportunity to revise this submission following peer review and for this thorough and very helpful analysis of our research work. After consultation with the other authors, we have reviewed the manuscript and performed the suggested improvements. Our point-by-point response to the Reviewer’s comments, suggestions and questions are described in detail below.

Reviewer 2: The authors presented an interesting study investigating the relationship between familial hypercholesterolemia and retinal microvascular changes assessed by optical coherence tomography angiography. The topic of the manuscript is original in content, and the findings are worth reporting, but the authors should improve the manuscript for English language and address the following comments before publication can be considered.

Abstract

Lines 19-21: please reformulate the sentence not as a question but in a more scientific way, such as “It is not known if an association exists between retinal microvascular changes and the presence of atheroma in patients with FH” or “the relationship between FH and retinal microvascular changes needs further investigation” or a similar statement to introduce the rationale from the study, but not in a form of a question

As suggested, we modified this sentence in the revised version of our manuscript as follows (line 19-21)

“The relationship between retinal microvascular changes and the presence of atheroma in patients with FH (FH group), and comparison to volunteers without FH (CT group) needs further investigation.”

Introduction

The manuscript should be revised and improved for English language: for example at line 14 “In the most frequent form, autosomal dominant heterozygous transmission, cardiovascular risk is increased 13-fold compared to the general population [4].” It should be replaced with “In the most frequent form characterized by/with an autosomal dominant heterozygous transmission, cardiovascular risk is increased 13-fold compared to the general population [4].”

We want to thank the Reviewer to give us the opportunity to improve our manuscript. We replaced this sentence in the revised manuscript (line 41-43)

“In the most frequent form, characterized by an autosomal dominant heterozygous transmission, cardiovascular risk is increased 13-fold compared to the general population”

Our manuscript was edited by a professional English medical writer in order to improve English language ([email protected] ; Isabella Athanassiou)

Before the paragraph related to OCTA, the authors should add some insight regarding FH and the eye. Has FH any known ocular clinical manifestations?

Thank you for raising this point. Indeed, the ocular changes in FH have been poorly studied in the literature. Corneal changes are known ocular manifestations of FH, with corneal arcus before 45 years old as a criterion in the Dutch Lipid Clinic Network score for the diagnosis of FH. But there are no described retinal changes in FH.

We added the following statement and reference in the revised version of our manuscript (line 57-60)

“High LDL-c levels in FH are associated with some ocular manifestations such as corneal arcus which is a diagnosis criterion for FH in the Dutch Lipid Clinic Network score when appears prior to 45 years [11]. However, no characteristic manifestations of hypercholesterolemia have been described at the retinal level.”

[11] Singh, S.; Bittner, V. Familial hypercholesterolemia--epidemiology, diagnosis, and screening. Curr Atheroscler Rep 2015, 17, 482-485, doi:10.1007/s11883-014-0482-5.

Methods

Line 97-98: “on objective refraction, intraocular pressure, axial length, slit-lamp examination results”: the authors should provide additional information about the instruments used to measure “objective refraction, intraocular pressure, axial length” and indicate the details of what they mean by “slit-lamp examination results”

We added additional information about the instruments used for this study, and a description of the slit-lamp examination.

Please find below the details as modified in the revised manuscript (line 105-112

“In addition to SS OCT-A, we collected information on objective refraction, intraocular pressure (Nidek Tonoref II autorefractokeratometer, Nidek Ltd, Gamagori, Japan), axial length (IOL Master 500, Carl Zeiss Meditec Inc, Dublin, USA), slit-lamp examination results, and ultra-widefield fundus imaging findings (Clarus 500, Carl Zeiss Meditec Inc., Dublin, USA and/or Optos California, Optos Inc., Malborough, USA). Slit lamp examination allowed us to analyze the anterior segment including the lens status. Slit lamp examination allowed us to analyze the anterior segment including the lens status and fundus photography was checked to assess the absence of retinal disease and optic neuropathy.”

Statistics: The author should provide a statistical power estimation for their study or at least some justification of the study n and add it to the “statistical analysis section

To our knowledge, this is the first study interested in FH and retinal vascular densities changes using Plexelite SS OCT-A. Unlike our study on cardiovascular disease and OCT-A (Arnould L et al. IOVS 2018), we had no previous data to estimate the vascular densities in both group and calculate the number of subjects needed to achieve a given statistical power. This study was a descriptive first pilot study. In that respect, we included consecutively all patients attending a cardiology consultation for FH follow-up between October 1, 2020 and May 31, 2021. The number of patients included is based on the capacity of recruitment of FH patients in the Cardiology department during their scheduled follow-up. Eighty-nine FH patients were screened, 2 patients refused to participate and 4 were excluded due to retinal disease or impossibility to achieve high quality SS OCT-A acquisitions (lack of fixation, advanced cataract). Volunteers constituting CT group were recruited during the same period in order to achieve a 1:1 ratio.

We added the following precisions in the revised manuscript:

Line 83-84:

« The number of patients included in this pilot study was based on the capacity of recruitment in the Cardiology department relying on the rhythm of FH patient’s follow-up. »

Line 87-89:

« Volunteers with no history of FH were recruited in the Ophthalmology department during the same period to constitute the control group (CT group) in order to achieve a 1:1 ratio. »

Line 154-156:

“Eighty-nine FH patients were screened for this study, 2 patients refused to participate and 4 were excluded due to retinal disease or impossibility to achieve high quality SS OCT-A acquisitions (lack of fixation, advanced cataract).”

Results

Line 142-143: “As presented in Table 1, the demographic characteristics of patients from the FH and CT groups were similar: please replace the term “similar” with a more objective and scientific term, or reformulate the sentence.

We reformulated this sentence with statistic terms (line 157-158)

“As presented in Table 1, the demographic characteristics of patients from the FH and CT groups did not show any statistically significant difference.”

When commenting about the differences between the two groups in terms of demographic characteristics, the authors should add in the description the indication of the statistical significance (p-value) of the differences, and not only the percentages.

Thank you for raising this point, this information was available in Table 1 but missing in the text. We added this in the revised manuscript (line 163-167)

“Systemic hypertension was more prevalent in the FH group than in the CT group (36.1% in the FH group versus 14.5% in CT group, P = 0.002), whereas the rate of diabetes was similar in both groups. FH patients had a higher rate of exposure to smoking than the CT patients did (43.4% of current or former smokers in the FH group vs. 18.7% in the CT group, P < 0.001).”

Discussion

All the limitations of the study should be presented in a single paragraph, and not in multiple ones at the end of the discussion

Many thanks to the Reviewer for pointing this out. We reorganized the limitations of the study in a single paragraph (line 298-327)

The authors should provide more insight about the potential practical applicability in the clinical practice of their findings

The clinical potential applicability of OCT-A in FH patients’ management is the ultimate goal to which this study represents a first step.

We added the following statement to the revised manuscript (line 244-246)

“In practical terms, a lowering in vascular densities could alert on atherosclerosis burden and lead to further cardiological investigation and/or adjustment of LDL-c lowering medication.”

Reviewer 3 Report

The paper entitled “Retinal microvascular changes in familial hypercholesterolemia: analysis with swept-source optical coherence tomography angiography” is a study based on the association between retinal vascular density with OCT and the presence of atherosclerosis in patients with familial hypercholesterolemia (FH). The manuscript is interesting, innovative, and of potential clinical interest.

The results show that The FH group had a significantly lower foveal avascular zone in OCT scans when compared to the control group. Retinal microvascularization and retinal vascular densities seem to be clinically different in the FH group when compared to normal subjects. One can expect to find vascular alterations in various organs and tissues in diseased patients, however, the early alterations detected in patients at risk with FH merits attention and can be of potential diagnostic interest in screening patients and managing select patients over time.

The study has been correctly planned. It is well written and of clinical interest. The study provides objective results, which adds to current literature in this field.

There are, however, several issues that need to be addressed by the authors, which include:

  1. The exclusion criteria include eyes with history of retinal disease. The authors should mention whether or not patients with glaucoma and/or neurologic disorders were also excluded, considering that these conditions can affect retinal nerve fiber layer thickness and retinal microvascularization. If not, these issues should be included as a limit to the study.
  2. Table 1 shows that the FH group had a mean age of 83 years as opposed to 78 years in the control group. Mention should be made as to whether this difference can be of clinical importance regarding retinal microvascularization. Further discussion on this issue warrants additional references.
  3. The authors raise an interesting point on page 9 regarding the potential use of OCT as a screening tool for patients with lipid anomalies. Further discussion on this point on how OCT could be implemented in a routine screenings setting could prove to be of clinical interest. Mention should be made as to whether or not OCT could be potentially used to manage patients with diagnosed altered microvascularization in determining treatment strategies to employ during follow-up visits.  
  4. The English can be improved for better flow.

Author Response

Manuscript ID jpm-1694763

“Retinal microvascular changes in familial hypercholesterolemia: 
analysis with swept-source optical coherence tomography angiography”

Thank you very much for this thorough and very helpful analysis of our research work and for all comments, suggestions and questions. We believe that all of Reviewers' comments are very precious and will contribute to improving the scientific and educational value of our article. Please find in detail below our point by point response to the suggestions and a revised version of the manuscript.

Reviewer 3: The paper entitled “Retinal microvascular changes in familial hypercholesterolemia: analysis with swept-source optical coherence tomography angiography” is a study based on the association between retinal vascular density with OCT and the presence of atherosclerosis in patients with familial hypercholesterolemia (FH). The manuscript is interesting, innovative, and of potential clinical interest.

The results show that The FH group had a significantly lower foveal avascular zone in OCT scans when compared to the control group. Retinal microvascularization and retinal vascular densities seem to be clinically different in the FH group when compared to normal subjects. One can expect to find vascular alterations in various organs and tissues in diseased patients, however, the early alterations detected in patients at risk with FH merits attention and can be of potential diagnostic interest in screening patients and managing select patients over time.

The study has been correctly planned. It is well written and of clinical interest. The study provides objective results, which adds to current literature in this field.

There are, however, several issues that need to be addressed by the authors, which include:

  1. The exclusion criteria include eyes with history of retinal disease. The authors should mention whether or not patients with glaucoma and/or neurologic disorders were also excluded, considering that these conditions can affect retinal nerve fiber layer thickness and retinal microvascularization. If not, these issues should be included as a limit to the study.

We thank the Reviewer for highlighting these important points.

In this study, we used the general term of “retinal disease” as a reference to macular disease, retinal periphery disease. Patients with a personal history of glaucoma, a current intraocular pressure lowering treatment or glaucoma suspect on the examination (optical nerve aspect on fundus photography and intraocular pressure) were excluded in both FH and CT groups.

We added this precision to the manuscript (line 92-93)

“The exclusion criteria were eyes with a history of retinal disease, optic neuropathy or a poor-quality image in SS OCT-A (signal strength of ≤ 7/10 or presence of artifacts due to eye movement or media opacities).”

However, general neurological disorders did not constitute an exclusion criterion. We agree with the concern raised by the Reviewer and added this bias to the limit of the revised manuscript (line 303-305)

“In addition to this, exclusion criteria were based only on the absence of ophthalmological disease while other general affections such as neurological disorders could also impact of retinal vascular densities [40].”

Added reference : [40] Augustin, A.J.; Atorf, J. The Value of Optical Coherence Tomography Angiography (OCT-A) in Neurological Diseases. Diagnostics (Basel) 2022, 12, 468-485, doi:10.3390/diagnostics12020468.

  1. Table 1 shows that the FH group had a mean age of 83 years as opposed to 78 years in the control group. Mention should be made as to whether this difference can be of clinical importance regarding retinal microvascularization. Further discussion on this issue warrants additional references.

The FH group had a median age of 56 years and the CT group had a median age of 49 years. We did not find a statistically significant difference between both groups for this feature. We agree that the age may influence vascular density parameters in OCT-A. Therefore, it was essential to ensure that this characteristic did not presents a statistically significant difference between FH and CT group. Moreover, to overcome any potential influence of age, we included this parameter in the multivariate analysis.

Following reference are linked to the effect of age on retinal microvascularization:

[34] Brücher, V.C.; Storp, J.J.; Eter, N.; Alnawaiseh, M. Optical coherence tomography angiography-derived flow density: a review of the influencing factors. Graefes Arch Clin Exp Ophthalmol 2020, 258, 701-710, doi:10.1007/s00417-019-04553-2.

Added reference: [36] Dastiridou, A.; Kassos, I.; Samouilidou, M.; Koutali, D.; Mataftsi, A.; Androudi, S.; Ziakas, N. Age and signal strength-related changes in vessel density in the choroid and the retina: an OCT angiography study of the macula and optic disc. Acta Ophthalmol 2021, doi:10.1111/aos.15028.

  1. The authors raise an interesting point on page 9 regarding the potential use of OCT as a screening tool for patients with lipid anomalies. Further discussion on this point on how OCT could be implemented in a routine screenings setting could prove to be of clinical interest. Mention should be made as to whether or not OCT could be potentially used to manage patients with diagnosed altered microvascularization in determining treatment strategies to employ during follow-up visits.  

Many thanks for raising this point concerning the clinical potential applicability of OCT-A in FH.

We believe that OCT-A could be potentially used to manage patients with FH by following the evolution of retinal microvascularization. We showed that lower densities are linked to higher CAC score. With further longitudinal investigation, we could imagine that during the follow-up of a given patient with FH, the occurrence of retinal densities lowering could be a sign of atherosclerosis burden, leading to treatment adjustment.

We added the following statement (line 244-246)

“In practical terms, a lowering in vascular densities could alert on atherosclerosis burden and lead to further cardiological investigation and/or adjustment of LDL-c lowering medication.”

Moreover, CAC score is an invasive examination which is not performed at each cardiological follow-up. Currently, the management of FH patients and treatment efficacy evaluation at each follow-up is based on blood LDL-c levels. This dosage does not reflect previous months LDL-c levels. Retinal vascular modification could reflect previous uncontrolled LDL-c levels.

We added this to the revised manuscript (line 257-263)

« Moreover, management of FH patients is currently based on the control of blood LDL-c levels. But LDL-c measure in a blood sample at a given time does not reflect previous months of cholesterol balance or imbalance. Yet, it is the chronic exposure to high LDL-c levels that leads to cardiovascular disease. Longitudinal study could provide further information about changes in retinal microvascularization over time and the potential memory of retinal vascular densities as a reflect of previous months’ LDL-c levels. »

  1. The English can be improved for better flow.

We want to thank the Reviewer to give us the opportunity to improve our manuscript. Our manuscript was edited by a professional English medical writer in order to improve English language ([email protected] ; Isabella Athanassiou)

Round 2

Reviewer 1 Report

All my comments have been well addressed. Thank you, 

Reviewer 2 Report

The authors addressed all my comments and in my opinion the manuscript can be accepted for publication in the current revised form.